# Chemical-Assisted Microwave Disinfection Used to Eradicate *Candida albicans* from Acrylic Resin Surfaces

**DOI:** 10.3390/jfb17010004

**Published:** 2025-12-20

**Authors:** Marek Witold Mazur, Anna Grudniak, Julia Konstancja Wawer, Dominika Gawlak

**Affiliations:** 1Department of Prosthodontics, Medical University of Warsaw, 02-097 Warsaw, Poland; dominika.gawlak@wum.edu.pl; 2Department of Bacterial Genetics, Institute of Microbiology, Faculty of Biology, University of Warsaw, 02-096 Warsaw, Poland; 3Student’s Research Group, Department of Prosthodontics, Medical University of Warsaw, 02-097 Warsaw, Poland; julia.k.wawer@gmail.com

**Keywords:** *Candida albicans*, microwaves, disinfection, biofilms, polymethyl methacrylate (PMMA), sodium hypochlorite (NaOCl), hydrogen peroxide (H_2_O_2_), chlorhexidine (CHX)

## Abstract

Microwave radiation is a potential alternative to conventional disinfection of acrylic resin, but exposure time must be minimized, e.g., by combining it with chemical agents, due to its effects on material properties. This study aimed to microbiologically evaluate the antifungal activity of microwave disinfection performed in distilled water, sodium hypochlorite (NaOCl), chlorhexidine (CHX), hydrogen peroxide (H_2_O_2_), or without immersion. Thermally polymerized PMMA samples colonized with *Candida albicans* ATCC 14053 were exposed to microwaves for 1 or 3 min in an unmodified microwave oven. Disinfection effectiveness was assessed by colony counting after 48 h of culture and absorbance after crystal violet staining. All microwave treatments significantly reduced fungal counts compared with the control (5360.00 ± 1663.09 CFU/mL). Complete inhibition of colony growth occurred only after 3 min exposure in distilled water, NaOCl, or CHX. One-minute exposure in these liquids reduced but did not eliminate fungi. The least effective method was disinfection without immersion, yielding 1040.00 ± 169.71 CFU/mL after 1 min and 560.00 ± 108.32 CFU/mL after 3 min. None of the tested conditions fully removed biofilms, although microwaves combined with NaOCl produced the best results. Overall, it was found that the presence of a liquid itself, rather than the type of chemical used, was the key factor in effective microwave-assisted disinfection. Microwave disinfection without the addition of chemicals does not remove biofilms.

## 1. Introduction

Acrylic resin, a popular material used for removable dentures, is characterised by the presence of microscopic spaces between supermolecules, which affects its adverse properties, such as water sorption and ease of colonisation by pathogenic microorganisms [1,2]. The most clinically important aetiological factor in fungal infections associated with dentures is *Candida albicans.* The mere placement of a denture in the oral cavity creates non-physiological conditions and, in consequence, provides a suitable environment for the development of infections [3]. In the population of removable dentures users, consisting mainly of older adults, their limited manual dexterity makes it difficult to maintain proper hygiene of the prostheses. In addition, reduced immunity, multimorbidity, and polypharmacy further disrupt the balance of the oral flora [1,3]. These conditions are conducive to infections, especially fungal, which pose a serious threat to the body not only locally but also systemically. Pathogens colonising dentures can contribute to the development of respiratory diseases, resulting in life-threatening complications [1,4].

The colonisation of the acrylic structure by pathogenic microorganisms increases the risk of cross-infection. The risk group includes primarily people in hospitals or nursing homes who, due to manual impairment, do not clean their dentures themselves but leave this task to the staff, who are also at risk of exposure. Another common source of cross-contamination is the prosthetic laboratory, where dentures from different individuals are sent and handled [2,4,5,6]. To prevent this complication, it is necessary to disinfect dentures effectively. According to Al-Makramani et al., recommended methods for disinfecting acrylic dentures in a prosthetic laboratory include ultrasonic cleaning, immersion in chemical disinfectants (e.g., 12 h in glutaraldehyde), and sterilization with ethylene oxide. All these methods require specialised equipment and reagents, which are not available in home conditions [6].

The solution proposed in the literature is the use of microwave radiation. The effectiveness against microorganisms results from the heating of the prostheses above the critical value of the proteins that they consist of, as well as non-thermal effects such as the impairment of cell metabolism directly by radiation, the heating of specific substances that absorb energy to a greater extent than water or the creation of pores as a result of interaction with the polarised cell membrane [7,8,9,10,11]. According to Karibasappa et al., a microwave oven cycle at 1350 W and 16 min provides similar efficiency to that of an autoclave. Furthermore, microwave sterilisation consumes 80% less energy than the use of an autoclave and takes much less time [12]. Studies on the disinfection of acrylic resin have shown that microwave disinfection produces clinical results comparable to those obtained by soaking acrylic resin in a chlorhexidine solution for 10 min [8]. In the case of fungal-induced stomatopathies, it can successfully replace an antibiotic course of treatment [7,13,14,15,16]. Neppelenbroek et al. found that the use of microwaves alone and in combination with the topical application of miconazole produced similar results in terms of reducing inflammation on the palate. The use of antifungal medication alone or specific oral hygiene instructions was not effective in reducing the clinical symptoms of denture-related stomatitis and the number of *Candida* colonies [7].

Numerous reports indicate that placing dentures in a domestic microwave oven for 3 min at 650 W allows the prosthetic restoration to be disinfected [8,16,17,18,19]. Despite its high antimicrobial efficacy, this method is not yet accepted in clinical practice. An undeniable advantage of the microwave method is the penetration of radiation into the inner layers of the material. *C. albicans* can penetrate up to 631 μm into the acrylic resin, where it is often unreachable by chemical compounds [15]. Studies have shown that microwave radiation leads to the polymerisation of residual monomer, which is associated with volume shrinkage [20,21,22]. There are also doubts regarding the immersion of the prostheses in liquids during microwave treatment. Indeed, this method allows for better temperature distribution in the material. It improves the ability to destroy microorganisms, as opposed to microwave treatment without immersion, which does not guarantee complete disinfection of the denture [23,24].

One method of limiting the destructive impact of microwave radiation on material is to reduce its exposure time. However, as it has been shown by research conducted by Senna et al. [11], shorter application times and lower power lead to a reduction in antimicrobial efficacy. One solution is to combine microwave and chemical disinfection. The use of two different types of decontamination techniques may result in maximized effectiveness and the reduction in undesirable effects. Research on this topic was conducted by Martínez-Serna et al. [25]. In their study, the use of both hydrogen peroxide and microwaves was an effective method of disinfecting acrylic resin. They concluded that “The efficacy of biofilm inhibition seems to be associated with the liquid in which the dentures are immersed in”.

The aim of this study was to evaluate in vitro the antifungal and anti-biofilm effectiveness of microwave and chemical-microwave disinfection using sodium hypochlorite (NaOCl), chlorhexidine (CHX), and hydrogen peroxide (H_2_O_2_) solutions. To date and according to our knowledge, the combination of microwave disinfection with chemical agents other than hydrogen peroxide or cleaning tablets has not been investigated so far. Therefore, this study fills the gap by analyzing the antifungal potential of new combinations of microwave and chemical disinfection.

## 2. Materials and Methods

### 2.1. Study Design and Preparation of Acrylic Samples

This in vitro experimental study evaluated the antifungal effectiveness of microwave and combined chemical-microwave disinfection methods for acrylic denture base resin. Acrylic samples contaminated with *Candida albicans* were treated using various disinfection protocols, and antifungal efficacy was determined by colony counting and the absorbance measurement at a wavelength of 570 nm (A570). The samples used in this study were made from thermally polymerised FuturaGen acrylic (Schutz Dental, Rosbach vor der Höhe, Germany). Using a laboratory scale (Mettler AT 200, Mettler-Toledo, Greifensee, Switzerland), 5 g of liquid was weighed in a silicone beaker, and then powder was added until a mass of 16.67 g was achieved. The amount of liquid to powder used was as stated in the manufacturer’s recommendations. The compounds were mixed by hand using a metal spatula and then, in the dough phase, placed between two smooth glass plates with 2 mm thick stoppers to ensure uniform thickness of the acrylic plate. The glass plates were manually pressed and secured with rubber bands, then placed in a Polyclav pressure vessel (Dentaurum, Ispringen, Germany) containing water heated to 45 °C and maintained at 2 bar for 20 min. After removal from the vessel, the glass plates were separated, and the plastic plates were removed, from which samples measuring 10 mm × 10 mm (+/−0.5 mm) were cut using a 0.5 mm thick carborundum disc. Eighty samples were prepared in this manner. The cut surfaces were sequentially treated with P40 and P80 sandpapers, followed by polishing with a rubber mounted on a straight handpiece. All sample surfaces were then roughened with P120 sandpaper. The finished samples were mixed and randomly selected in batches of five. Each batch was packaged separately in sterilization sleeves and sterilized with ethylene oxide under controlled conditions. The process was conducted at 55 °C, with 59% relative humidity, for a total of one hour. After sterilization, the samples were allowed to degas for 12 h to ensure that any remaining ethylene oxide was fully removed.

### 2.2. Preparation of Biological Samples for Testing

Sterile samples prepared as described above were immersed in a rejuvenated culture of *Candida albicans* ATCC 14053 with a density of 0.5 on the McFarland scale, obtained from the collection of the Institute of Microbiology, University of Warsaw. The fungi were cultured in Sabouraud Medium (Difco^TM^, Pinellas Park, FL, USA) at 37 °C for 48 h without shaking. Throughout the duration of the fungus culture, a biofilm of *C. albicans* formed on the plates. After 48 h of cultivation, the samples were removed with sterile tweezers, rinsed in sterile water, and dried on sterile Whatman paper. The acrylic samples with *C. albicans* biofilm were then subjected to disinfection.

### 2.3. Microwave Disinfection Assisted by Chemical Compounds

In the experiment, distilled water was used and commercially available solutions of 2% chlorhexidine (Glucochexin 2%, Cerkamed, Stalowa Wola, Poland) and 3% hydrogen peroxide (Farmina, Cracow, Poland). Due to the inaccessibility of a ready-made 0.5% sodium hypochlorite solution, it was decided to dilute 2.5 mL of 2% solution (Chloraxid 2%, Cerkamed, Stalowa Wola, Poland) with 7.5 mL of distilled water. However, the final concentration of the prepared solution was not verified by measurement. Acrylic plates bearing *C. albicans* biofilms on their surfaces were randomly divided into groups detailed in Table 1.

For the assessment of *C. albicans* colony counts, four independent trials were conducted. The sample size was determined statistically using G*Power 3.1.9.7 [26], with: α = 0.05, power = 0.95, and the effect size = 1. Effect size was estimated based on the means and standard deviations reported in previously published studies [8,19,27]. For the absorbance measurements, three independent trials were performed, the amount of the formed biofilm was determined after staining with crystal violet, according to the procedure described by O’Toole and Kolter (1998) [28].

Ten millilitres of the test solution were poured into prepared sterile glass tubes according to the previous allocation, i.e., distilled water, 0.5% sodium hypochlorite, 2% chlorhexidine, and 3% hydrogen peroxide. In Dry-1 and Dry-3, no solutions were added to glass test tubes, and an additional control group of the entire experiment was provided by a sample treated with neither chemicals nor microwaves. An acrylic plate with biofilm deposited on it was randomly placed in each of the prepared test tubes. Four test tubes containing liquids were placed in a beaker containing 160 mL of tap water, resulting in a total liquid volume of 200 mL in the beaker. Then, using a digital thermometer (Shenzhen Yixing Micro Technology, Shenzhen, China), the temperature of the water in the vessel was checked and found to be 25 °C (±2 °C). The measurement was taken at the center of the container, by immersing the sensor in the space between the test tubes. Sample beakers were placed on the microwave oven turntable, and a test tube without liquid was placed next to them (Figure 1). Microwave disinfection was carried out in an unmodified domestic microwave oven (MWP 252 SB, Whirlpool^TM^, Benton Harbor, MI, USA) set at 650 W. Depending on the variant of the experiment, the oven was set for 1 or 3 min. After the microwave oven had finished its course, the beaker was removed from it, and the water temperature was measured as described above. The difference between the initial and final temperatures was recorded to estimate the amount of energy supplied by the oven in each operating cycle. In the control group, the samples were not subjected to modifications.

### 2.4. Quantitatfication of C. albicans After Disinfection

After disinfection, acrylic resin samples were removed from the test tubes using sterile tweezers, gently rinsed with sterile distilled water, and then placed in 10 mL of sterile saline solution. The samples used as a control for the growth of *C. albicans,* which were not subjected to disinfection, were transferred aseptically to 10 mL of saline. To remove the *C. albicans* biofilm from the acrylic surface, all samples were shaken using a Vortex device (JWE ML-962, JWE Electronic, Warsaw, Poland) for 1 min at a speed of 3000 rpm. After shaking, 25 µL of the obtained solution was transferred to a Petri dish with Sabouraud agar using an automatic micropipette and incubated for 48 h. The *C. albicans* titre was quantified on agar plates postincubation by colony counting and subsequent calculation of CFU/mL. Four repetitions were performed in each group.

### 2.5. Absorbance Measurement

The remaining 3 samples in each group (33 samples) were subjected to absorbance testing. Biofilm formation was quantified using the crystal violet (CV) staining method described by O’Toole and Kolter (1998) [28]. Samples after disinfection were removed aseptically from test tubes, rinsed twice in sterile distilled water and stained with 2 mL of 0.1% CV for 10 min at room temperature. The working solution was prepared by diluting the CV stock (0.8 g CV, 20 mL ethanol, 0.8 g ammonium oxalate, 80 mL water) to 0.1%. After staining, samples were rinsed twice with distilled water, transferred to a fresh 24-well plate, and dried at 50 °C for 15 min. The retained dye was solubilized with 1 mL of ethanol for 15 min with shaking. The resulting solution was transferred to a clean titration plate, and absorbance was measured at 570 nm using a Sunrise plate reader (Tecan, Männedorf, Switzerland).

### 2.6. Statistical Analysis and Use of Gen AI Tools

The obtained results were statistically analysed using IBM SPSS Statistics 29.0.2.0 (20) software (IBM, Armonk, NY, USA). Descriptive statistical analysis was performed (Table 2 and Table 3). The normality of data distribution in subgroups was checked using the Shapiro–Wilk test supported by visual inspection of Q–Q plots. For the colony count measurements, two groups showed non-normal distributions; therefore, a non-parametric Kruskal–Wallis one-way analysis of variance with step-down multiple comparisons was performed. On this basis, homogeneous subsets designated by letters in Table 2 were identified. In the case of absorbance measurements, due to the small sample size, greater emphasis was placed on graphical inspection of Q–Q plots. The data were normally distributed; therefore, ANOVA was performed, but no statistically significant differences between groups were observed (*p* = 0.135). The colony counts and absorbance measurements were performed on different sets of samples; therefore, no correlation analysis between the two variables was conducted.

During the preparation of this manuscript, the authors utilized ChatGPT (GPT-5-mini, accessed in 2025, OpenAI, San Francisco, CA, USA) for the identification of the relevant scientific literature, supporting the selection of appropriate statistical methods, assistance in language editing, and translation of text. Additionally, EndNote (Clarivate, Philadelphia, PA, USA) was used for the management and formatting of citations. The authors checked and improved all results obtained using AI.

## 3. Results

This study showed that microwave disinfection is effective against *C. albicans* ATCC 14053. Each group that was exposed to microwave radiation and then incubated for 48 h showed a significantly lower number of characteristic fungal colonies compared to the control group, as shown in Figure 2 and in the photographs of Petri dishes showing the cultured colonies (Figure 3). The highest efficiency was achieved by 3 min disinfection in groups: Aq. Dest.-3, NaOCl-3 and CHX-3. In these groups, no growth of fungal colonies was observed in 4 trials. In group H_2_O_2_-3, the results were slightly worse, with 120 CFU/mL achieved in one of the repetitions. The highest *C. albicans* titre among the groups subjected to microwave treatment for 3 min was observed in the dry disinfection. The average CFU/mL level was 560 and was shown to be higher than in the other groups, and the difference was statistically significant.

Microwave disinfection for 1 min proved to be less effective than for 3 min in all trials. In any of the groups, complete sterility was not achieved. In both 1 min and 3 min disinfection, immersion in hydrogen peroxide resulted in higher *Candida* counts compared to other disinfectants. However, the comparison between groups using hydrogen peroxide and distilled water provided different results. The average CFU/mL was higher in Aq. Dest.-1 group than in H_2_O_2_-1; unlike in the case of disinfection carried out for 3 min (Table 2). The lack of sample immersion gave results several times higher than those of acrylic resin immersed in liquid. As in the case of groups disinfected for 3 min, immersion of the material in the liquid gave significantly better antifungal results in the statistical analysis.

The results obtained after staining the biofilm mass using the classical crystal violet method showed absorbance values ranging from 0.085 in the NaOCl-3 group to 0.165 in the control group (Table 3). Statistical analysis revealed no significant differences between the groups. In contrast to the quantitative CFU measurements, where the lowest values after 1 min exposure were recorded in the CHX-1 group, the absorbance data indicated that the most pronounced reduction in biofilm mass occurred in the sodium hypochlorite treated samples, and this effect was independent of microwave exposure time. For the 3 min disinfection protocol, all groups exhibited lower absorbance values compared with the corresponding groups exposed to microwaves for 1 min. Additionally, groups treated with antimicrobial agents showed reduced biofilm absorbance relative to groups that did not contain antimicrobial components (Figure 4).

During microwave operation, the liquid containing the test tubes reached an average temperature of 59.53 °C (SD = 2.21 °C) during the 1 min cycle and an average of 91.2 °C (SD = 2.22 °C) in the 3 min treatment course.

## 4. Discussion

In times of widespread antibiotic overuse, increased resistance to chemotherapeutics, and a growing population of older adults using acrylic dentures, it is crucial to find quick, effective, and cost-effective methods of disinfecting prosthetic restorations. Combating opportunistic pathogens such as *Candida albicans,* which pose a serious threat to immunosuppressed patients, is considered particularly important [1,3,4]. By conducting this research, the authors addressed this issue by exploring the potential use of microwave disinfection as an alternative to conventional techniques to decontaminate prosthetic restorations.

The principal limitation that prevents the implementation of microwave disinfection in clinical practice is the insufficiency in the extensive research on the effects of microwave radiation on the acrylic resin properties, especially polymerization shrinkage, coupled with thermal deformation of the material [20,21,22]. The solution proposed by the authors is to immerse the material in disinfectants during microwave radiation treatment, which can result in maintaining antifungal efficacy in the shortest possible exposure of the prosthetic restoration to the harmful effects of microwaves and maximally reducing the amount of time that the material is subjected to elevated temperatures. Research in this field was undertaken by Senna et al. [29]. They studied the combination of microwave disinfection for 1, 2, and 3 min with enzymatic tablets for cleaning dentures contaminated with *C. albicans*. No visible microorganisms were observed when microwave radiation was used with distilled water for 3 min at 450 W. When microwaves were combined with cleaning tablets, only 2 min at the same power was sufficient to achieve the absence of any cells. Moreover, a 2 min exposition with the addition of cleaning tablets resulted in the lowest temperature increase among all the groups in which no cells were achieved. A similar study by Choudhry et al. performed on PMMA infected with bacteria showed analogous results [30]. This confirms that the combination of the two disinfection methods is an effective way of reducing the microwave treatment time.

The results obtained in this study indicate a significant reduction in *C. albicans* counts after exposure to microwave radiation. Combining chemical methods and microwave radiation increases the effectiveness of antifungals. Regardless of exposure time (1 or 3 min), immersion in liquid significantly improved the antifungal effect of microwave disinfection compared to dry conditions. The differences in effectiveness among individual disinfectant solutions were minimal, suggesting that the critical parameter for microwave disinfection is the presence of a liquid medium, with the specific formulation of the disinfectant playing a comparatively minor role. These results are consistent with data reported in the literature, where immersion in a liquid containing water is considered an essential and integral factor that allows for microwave disinfection [23,24]. A study by Weeb et al. showed that for acrylic samples, whether contaminated with *C. albicans* or with *S. gordonii*, up to 6 min of microwave operation at 350 W was required for sterilization when not immersed [31]. A study conducted by Dixon et al., although it concerned relining materials, clearly showed that the use of microwaves under dry conditions does not lead to complete sterilisation of *C. albicans*-contaminated samples [24]. However, contrasting conclusions were reached by Robati Anaraki et al. [32]. In their study on gypsum models, which employed the highly resistant bacterium *B. subtilis*, no differences in efficacy were observed between disinfecting dry and moist models. The use of this particularly resilient species may explain their divergent results. The problems with disinfection without immersion in liquid can be attributed to the formation of the so-called “cold spots”. As shown by the experiments of Rohrer and Bullard, radiation in a microwave oven is not evenly distributed throughout the space. This leads to the formation of “cold spots” that do not absorb a sufficiently high dose of radiation and, as a result, are not adequately sterilised [33].

Despite the strong antimicrobial activity of microwave irradiation, it proved ineffective in removing biofilms from the acrylic surface. To assess this, a well-established method for visualizing biofilms was utilized- absorbance measurements after crystal violet staining. Crystal violet stains not only fungal cells but also the extracellular matrix. The method does not evaluate the microbiocidal effectiveness of a disinfection protocol; rather, it reflects the extent to which biofilm structures are removed during decontamination [34,35,36].

The results obtained in this experiment clearly demonstrate that microwave disinfection is not an appropriate method for biofilm removal. The absorbance values confirm the presence of biofilm on the material surface in all experimental groups, regardless of the antimicrobial effectiveness previously demonstrated by the colony count analysis (Figure 4 and Figure 5). Similar conclusions were reported by Sesma et al., who examined contaminated samples using SEM. Microwave disinfection with brushing resulted in significantly higher levels of observed biofilm than the combined approach of microwaves, brushing, and a chemical cleaning agent. Moreover, SEM images revealed morphological changes, including compromised cell walls, although the microorganisms still adhered to the surface [37].

Disinfection using microwave irradiation in combination with sodium hypochlorite or chlorhexidine appears to be the most effective approach for reducing biofilm presence. In both conditions, the biofilm is partially removed from the surface of the disinfected material, and the cells are dead, as confirmed by the colony count analysis. This effect is attributable to the chemical properties of both agents. Sodium hypochlorite has long been successfully used as a disinfectant not only because of its strong antimicrobial activity but also due to its lytic action, which is crucial for biofilm removal. Chlorhexidine can also infiltrate mature biofilm structures and directly harm fungal cells, resulting in the disruption of the biofilm matrix [38,39,40]. This is consistent with the findings of Ardakani et al., who examined acrylic materials infected with a polymicrobial biofilm. In their research, sodium hypochlorite achieved the most substantial decrease in biofilm mass, followed by chlorhexidine, whereas photodynamic therapy and laser irradiation produced comparatively weaker outcomes [41].

The effectiveness of H_2_O_2_ in biofilm removal was only slightly greater than that of distilled water. However, its fungicidal activity, when combined with microwave irradiation was quite high, as confirmed by the earlier colony count analyses. Unlike distilled water, hydrogen peroxide possesses the ability to disrupt *C. albicans* biofilm structures [34]. Nevertheless, in our study, it was subjected to microwave heating, which likely caused thermal decomposition and a reduction in its activity [42]. Microwave disinfection demonstrates only partial activity in the removal of *C. albicans* biofilms and is unable to eliminate them effectively. Therefore, meticulous mechanical cleaning of the denture prior to disinfection is recommended [37,43].

Immersing the samples in distilled water produced notable results. Although it lacks intrinsic antimicrobial activity, when combined with microwave radiation, it demonstrates an antifungal effectiveness comparable to that of disinfecting solutions. This effect is likely due to its hypoosmotic nature relative to the microbial cytoplasm. Due to osmotic pressure, distilled water penetrates the interior of microorganisms, increasing the water content in cells [44]. Therefore, more distilled water (H_2_O) molecules can be excited by microwave radiation, which improves the overall effectiveness of disinfection.

Compelling observations were made when comparing groups containing hydrogen peroxide as a liquid. Despite a 3 min exposure, the H_2_O_2_-3 group showed incomplete disinfection. Group H_2_O_2_-1 was also less effective than other liquid disinfectants, though superior to distilled water. One possible explanation for this phenomenon is the disruption of microwave transmission by bubbles formed on the surface of a sample (Figure 6). These bubbles are probably composed with oxygen as a result of the thermal decomposition of hydrogen peroxide at increased temperature and steaming water [42,45]. The oxygen formed on the surface of the sample is not a dipole and therefore cannot be effectively heated by microwave radiation. Generally gases have low dielectric loss factors and low molecular density compared to liquid water, which means they absorb microwave energy very inefficiently [10,46,47,48]. As a result, these bubbles act as insulators, leading to uneven heating and consequently reduced disinfection effectiveness. When microwaves are applied for 1 min, the increase in temperature is low; so, hydrogen peroxide is probably only slightly decomposed and retains its antimicrobial effectiveness, but during 3 min exposition, the temperature reached more than 90 °C. Similar results of combining microwaves with hydrogen peroxide were obtained by Martinez-Sereena et al. [25]. This study examined the antimicrobial effectiveness of combining microwave disinfection with distilled water and hydrogen peroxide against *C. albicans* biofilm on polished and rough acrylic resin. The optimal agent for microwave disinfection was surface-dependent, with hydrogen peroxide slightly superior for polished surfaces and distilled water for rough ones [25]. A possible explanation of this phenomenon is better adhesion of gas bubbles to a rough surface, but further research in this field is needed.

This study has certain limitations, including the small number of samples per group and the use of a single microwave exposure only with a power of 650 W, which does not fully represent clinical conditions. The effects of microwave disinfection on PMMA physical properties, such as colour, surface roughness, and mechanical strength, were also not evaluated. Moreover, only *Candida albicans* was tested, while denture biofilm contains different microbes with different susceptibility to microwaves [27]. Future studies should include larger sample sizes, assess material changes after repeated disinfection with different microwave power, and evaluate the effectiveness of these methods against mixed microbial biofilms.

## 5. Conclusions

Microwave radiation, regardless of the disinfection protocol used, proved to be an effective method to reduce *C. albicans* counts. The protocol, consisting of 3 min of exposure with immersion of the material in liquid, allows for complete disinfection of the material. Each test sample had a significantly lower fungal colony count compared to the control sample. Contrary to expectations, the combination of microwave disinfection and immersion in a chemical disinfectant did not meaningfully improve the effectiveness of the process compared to the use of pure distilled water. The use of chemical agents with lytic activity such as sodium hypochlorite results not only in disinfection but also in partial removal of biofilm structures from the acrylic surface. Prolonged exposure to microwave radiation at 650 W reduces the disinfectant properties of hydrogen peroxide.

## Figures and Tables

**Figure 1 jfb-17-00004-f001:**
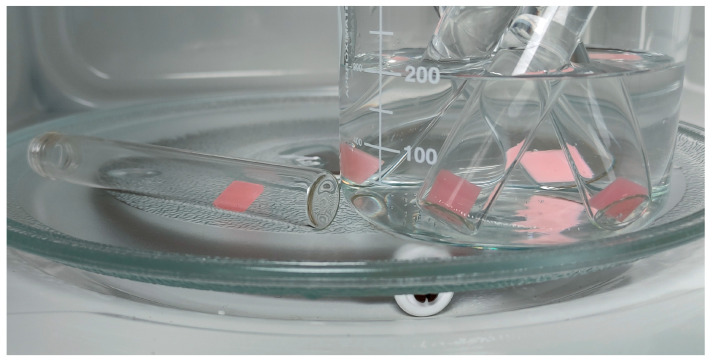
Samples in test tubes placed on the turntable of microwave oven before disinfection.

**Figure 2 jfb-17-00004-f002:**
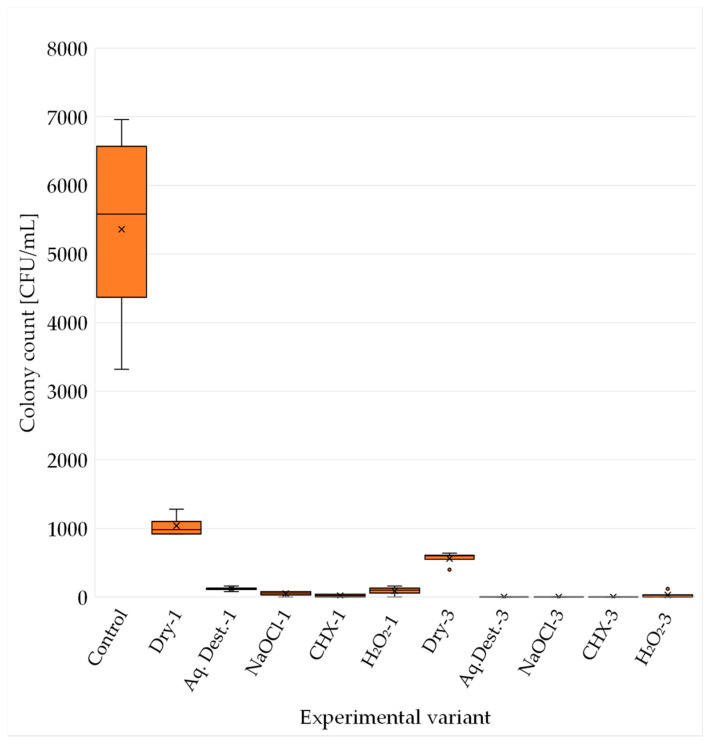
*C. albicans* colony count (CFU/mL) after 48 h incubation for each experimental variant. The mean is marked with an X and the median with a horizontal line. Quartiles were calculated using the inclusive method (including the median). The poorer efficiency is clearly visible in case of microwaves without immersion (Dry-1 and Dry-3).

**Figure 3 jfb-17-00004-f003:**
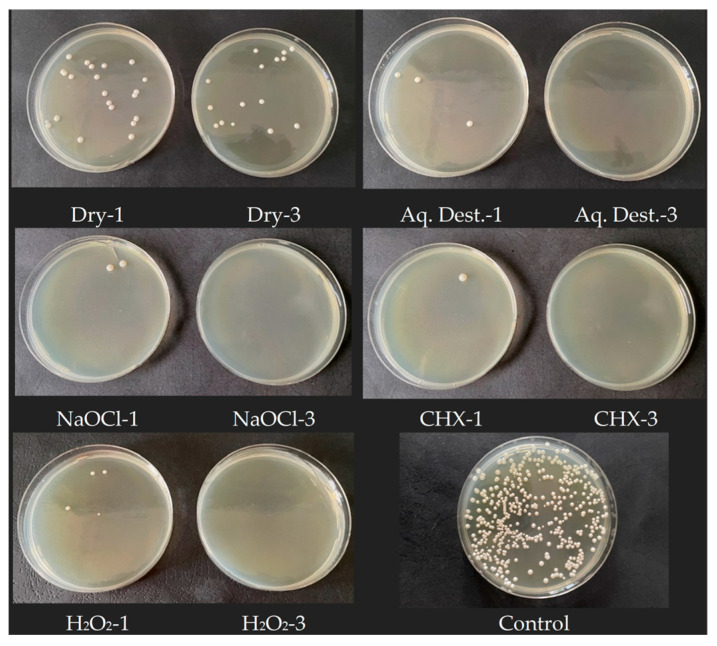
Sample photos of Petri dishes with visible number of *C. albicans* colonies. 1 min microwave exposition is presented on left and 3 min on right. The higher efficiency is clearly visible after 3 min of exposure to microwaves in each experimental variant.

**Figure 4 jfb-17-00004-f004:**
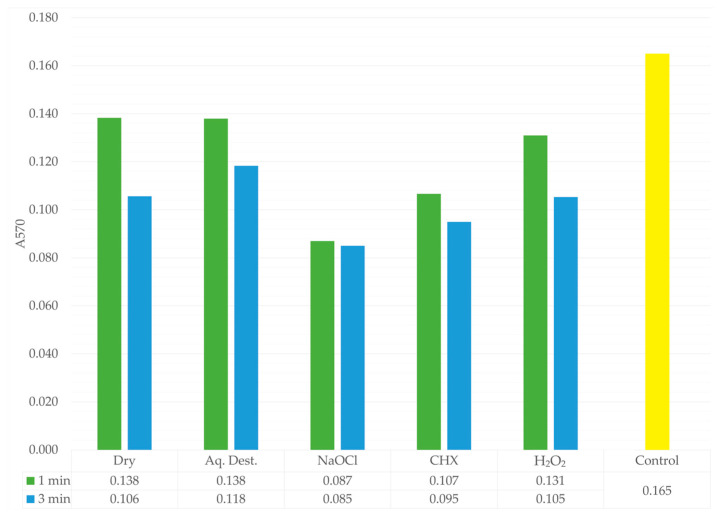
Absorbance values of the biofilm after crystal violet staining for each experimental group. 1 min is shown in green, 3 min in blue, and the control group in yellow. A pronounced reduction in absorbance is observed in the NaOCl group, indicating its strong lytic activity and effective degradation of the biofilm mass. The lower table presents the mean absorbance values for each group.

**Figure 5 jfb-17-00004-f005:**
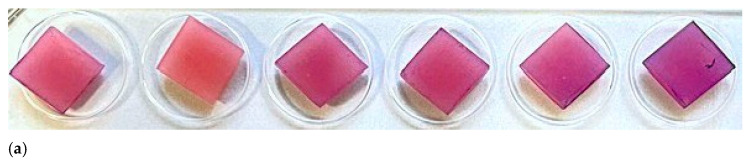
Samples after crystal violet staining on a 24-well plate. (**a**) Samples after 1 min microwave disinfection, from left: NaOCl-1, CHX-1, H_2_O_2_-1, Aq. Dest.-1, Dry-1, Control; (**b**) Samples after 3 min microwave disinfection, from left: NaOCl-3, CHX-3, H_2_O_2_-3, Aq. Dest.-3, Dry-3, Control.

**Figure 6 jfb-17-00004-f006:**
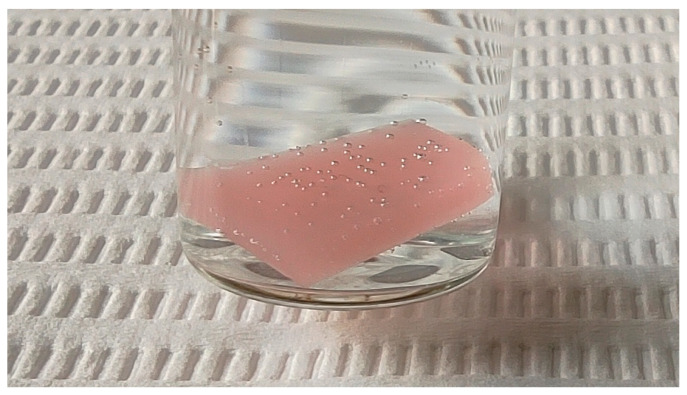
Gas bubbles formed on the surface of the sample.

**Table 1 jfb-17-00004-t001:** Characteristics of groups.

Experimental Variant	Microwave Operation Time (min)	Number of Samples (N)	The Liquid in Which the Samples Were Immersed
Control	0	7	None
Dry-1	1	7	None
Aq. Dest.-1	1	7	10 mL distilled water
NaOCl-1	1	7	10 mL 0.5% sodium hypochlorite
CHX-1	1	7	10 mL 2% chlorhexidine
H_2_O_2_-1	1	7	10 mL 3% hydrogen peroxide
Dry-3	3	7	None
Aq. Dest.-3	3	7	10 mL distilled water
NaOCl-3	3	7	10 mL 0.5% sodium hypochlorite
CHX-3	3	7	10 mL 2% chlorhexidine
H_2_O_2_-3	3	7	10 mL 3% hydrogen peroxide

**Table 2 jfb-17-00004-t002:** Mean, standard deviation, minimum, and maximum of *Candida albicans* colony counts after 48 h incubation. Different letters in superscript indicate statistically significant differences among groups based on the Kruskal–Wallis one-way analysis of variance with step-down multiple comparisons; groups sharing the same letter are not significantly different.

Experimental Variant	Mean (CFU/mL)	Standard Deviation (SD)	Minimum	Maximum
Control	5360.00 ^A^	1663.09	3320.00	6960.00
Dry-1	1040.00 ^B^	169.71	920.00	1280.00
Aq. Dest.-1	120.00 ^C^	32.66	80.00	160.00
NaOCl-1	50.00 ^CDE^	38.30	0.00	80.00
CHX-1	20.00 ^CDE^	23.09	0.00	40.00
H_2_O_2_-1	90.00 ^CD^	68.31	0.00	160.00
Dry-3	560.00 ^F^	108.32	400.00	640.00
Aq. Dest.-3	0.00 ^E^	0.00	0.00	0.00
NaOCl-3	0.00 ^DE^	0.00	0.00	0.00
CHX-3	0.00 ^DE^	0.00	0.00	0.00
H_2_O_2_-3	30.00 ^CDE^	60.00	0.00	120.00

**Table 3 jfb-17-00004-t003:** Mean, standard deviation, minimum, and maximum of absorbance at 570 nm (A570) of *C. albicans* biofilm after crystal violet staining. No statistically significant groups were found.

Experimental Variant	Mean	Standard Deviation (SD)	Minimum	Maximum
Control	0.165	0.038	0.123	0.196
Dry-1	0.138	0.036	0.101	0.172
Aq. Dest.-1	0.138	0.051	0.088	0.190
NaOCl-1	0.087	0.010	0.080	0.099
CHX-1	0.107	0.039	0.062	0.130
H_2_O_2_-1	0.131	0.049	0.082	0.180
Dry-3	0.106	0.025	0.077	0.125
Aq. Dest.-3	0.118	0.017	0.100	0.133
NaOCl-3	0.085	0.012	0.071	0.094
CHX-3	0.095	0.028	0.066	0.122
H_2_O_2_-3	0.105	0.024	0.078	0.121

## Data Availability

The original data presented in this study are openly available in FigShare at: https://doi.org/10.6084/m9.figshare.29949002.v2.

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
