# Peer review of "Chemical-Assisted Microwave Disinfection Used to Eradicate Candida albicans from Acrylic Resin Surfaces"

_jfb, 2025, doi:10.3390/jfb17010004_

Round 1

Reviewer 1 Report

Comments and Suggestions for Authors

An interesting topic with a significant workload!. It is well written and provides solid review of the relevant literature. Methodologically, the article is well-structured. It makes a significant contribution to the study of microwave disinfection of acrylic dentures.

Author Response

We would like to express our sincere gratitude for valuable comments and constructive suggestions. 

Comment 1: An interesting topic with a significant workload! It is well written and provides solid review of the relevant literature. Methodologically, the article is well-structured. It makes a significant contribution to the study of microwave disinfection of acrylic dentures.

Response 1: We sincerely thank the Reviewer for the positive evaluation of our work. No changes were necessary in response to this comment.

Reviewer 2 Report

Comments and Suggestions for Authors

Dear authors,I would like to express my gratitude for your research contributions. While the article is generally well-written, some additions would be appropriate. Firstly, it is essential to mention the study design in the introduction to the materials and methods section. Moreover, it is imperative that the study provides a detailed explanation of the methodology employed to determine the sample size, including the calculation of the sample size. In the discussion section, when comparing the results with those from other studies, it is essential to specify which microorganisms were used in the other studies, and the results of the current study should be compared accordingly.

Author Response

We would like to express our sincere gratitude for valuable comments and constructive suggestions. All remarks have been carefully considered, and the manuscript has been revised accordingly. Below, we provide a detailed response:

Comment 1: It is essential to mention the study design in the introduction to the materials and methods section.

Response 1: We have added a clear statement of the study design at the beginning of the Materials and Methods section (page 3, lines 110-114).

Comment 2: Provide a detailed explanation of the methodology employed to determine the sample size, including the calculation of the sample size.

Response 2: A detailed description of the sample size calculation and methodology has been added (page 4, lines 151-154). The calculation was performed using G*Power software, based on effect size values calculated from comparable studies. Considering that the differences between control and experimental groups in similar works are typically very large (effect size >2), the statistical power analysis indicated that total sample size should be 22 samples. To ensure higher robustness and reproducibility, we decided to use effect size = 1, which resulted in 44 samples.

Comment 3: In the discussion section, when comparing the results with those from other studies, it is essential to specify which microorganisms were used in the other studies, and the results of the current study should be compared accordingly.

Response 3: We have revised the Discussion section to indicate the specific microorganisms examined in previous studies and to align the comparison accordingly (page 8, lines 262-263, 268, 281-282, 285, 287; page 9, lines 318).

Reviewer 3 Report

Comments and Suggestions for Authors

General assessment:
This study investigates the application of microwave and chemo-microwave techniques for disinfecting PMMA colonised by Candida albicans. The subject is clinically significant, as infections associated with dentures continue to be a prevalent issue in prosthodontics. Overall, the manuscript is well-organised, with a clear method and good use of statistics. But there are a few things that could make it less likely to be accepted by a high-impact journal in its current form.

Limitations:
1. Originality:  the authors, Neppelenbroek, Silva, and da Costa have already looked into using microwaves to clean dentures. This paper presents the concept of integrating microwave radiation with chemical agents; however, the primary conclusion—that immersion is more critical than the specific disinfectant type—does not signify a significant innovation.
2. Study degign:
a. The small sample size (four specimens per group) weakens the validity of the statistical conclusions.
b. Also, the experiments only looked at one short exposure cycle, which isn't how things are usually done in the clinic.
c. Another important point is that there hasn't been any research on how the procedure changes the material itself. Factors such as the surface roughness, mechanical properties, or colour stability of PMMA are highly relevant for clinical acceptance but were not investigated here.
d. Additionally, testing solely C. albicans offers a limited perspective, as authentic denture biofilms typically encompass various microbial species.
3. Finally, the part about using AI needs to be rewritten so that it is clearer and more open. A better way to say this might be: "AI-assisted tools were only used for editing language and formatting references; the authors were responsible for all scientific content, data analysis, and interpretation."

Author Response

We would like to express our sincere gratitude for valuable comments and constructive suggestions. All remarks have been carefully considered, and the manuscript has been revised accordingly.

Comment 1:  The authors, Neppelenbroek, Silva, and da Costa have already looked into using microwaves to clean dentures. This paper presents the concept of integrating microwave radiation with chemical agents; however, the primary conclusion that immersion is more critical than the specific disinfectant type does not signify a significant innovation.
Response 1: We acknowledge that the antimicrobial use of microwaves for denture disinfection has been previously reported. However, since microwave exposure can negatively affect denture materials, minimizing exposure time is crucial for clinical safety. One potential way to achieve this is by combining microwaves with chemical agents. Our results confirmed that such a combination can be effective (e.g., after 1 minute of exposure, the CHX group showed six times lower CFU/mL than the Aq. Dest. group), although the effect was smaller than expected. The most pronounced differences were observed between groups with and without immersion, which explains our main conclusion. To our knowledge, this is the first study to evaluate such a wide range of chemical compounds in combination with microwave exposure, and further research is needed to identify the most effective and safe parameters

Comment 2a: The small sample size (four specimens per group) weakens the validity of the statistical conclusions.
Response 2a: A statistical calculation was performed using G*Power software. Due to the very large effect size (>2) typically observed in this field, four specimens per group were deemed sufficient for valid analysis. This justification has been included in the revised Matherial and Method section (page 4, lines 151-154).

Comment 2b: Also, the experiments only looked at one short exposure cycle, which isn't how things are usually done in the clinic.
Response 2b:We agree that clinical procedures often involve multiple cycles. This has been acknowledged as a limitation, and a suggestion for future studies was added (page 9, lines 323-330). However, it is important to note that in cases where dentures are disinfected in e.g., a dental laboratory, the process typically consists of a single disinfection cycle before further handling or adjustment. From a microbiological and safety perspective, the critical point is that the denture should be sterile after the first cycle, allowing it to be safely manipulated or used.

Comment 2c:  Another important point is that there hasn't been any research on how the procedure changes the material itself. Factors such as the surface roughness, mechanical properties, or colour stability of PMMA are highly relevant for clinical acceptance but were not investigated here.
Response 2c: We agree that the effect of disinfection procedures on PMMA material properties is clinically very important. This has been acknowledged as a limitation, and a suggestion for future studies was added (page 9, lines 323-330).

Comment 2d: Additionally, testing solely C. albicans offers a limited perspective, as authentic denture biofilms typically encompass various microbial species.
Response 2d:  Infections with Candida albicans are among the main causes of denture stomatopathy leading even to denture replacement, what results in additional patient costs. We also highlighted in manuscript that fungal infections, are particularly challenging to eradicate, as C. albicans is capable of penetrating deep into the acrylic surface, making its removal more difficult. In addition, there are few antifungal antibiotics as opposed to antibacterial ones. Microwave disinfection is based on physical factors and works similarly for all cells, but this factor was included in the revised version of the manuscript in limitation part (page 9, lines 323-330).

Comment 3:  Finally, the part about using AI needs to be rewritten so that it is clearer and more open. A better way to say this might be: "AI-assisted tools were only used for editing language and formatting references; the authors were responsible for all scientific content, data analysis, and interpretation."

Response 3: The statement regarding AI use has been rewritten for transparency. We want to reliably highlight how AI was used for our research. We clarified that:

“During the preparation of this manuscript, the authors utilized ChatGPT (GPT-5, OpenAI, USA) for identification of relevant scientific literature, support the selection of appropriate statistical methods , assistance in language editing and translation of text. Additionally, EndNote (Clarivate, USA) was used for the management and formatting of citations. The authors checked and improved all results obtained using AI.”

Reviewer 4 Report

Comments and Suggestions for Authors

The study addresses an important clinical and laboratory issue: optimizing denture disinfection methods using microwave and chemical-microwave combinations against Candida albicans.  However, substantial revision and clarification are necessary before the manuscript can be considered for publication.

  1. Title

Issue:
The title is long and repetitive. I Suggest simplifying it:

  1. Abstract

Issue:
The abstract is verbose and lacks numerical outcomes.
Suggestion:
Include main quantitative findings (e.g., “3-minute microwave + CHX achieved 0 CFU/mL versus 5360 ± 1663 CFU/mL in control”).
Remove redundant phrases and focus on result significance and conclusion.

  1. Introduction

Issue:
Well-referenced but overly descriptive; lacks a clear articulation of the research gap.
Add a concise final paragraph stating the precise knowledge gap addressed

  1. Materials and Methods

Issues and Suggestions:

  • Randomization: Specify how samples were randomly assigned to groups (e.g., computer-generated or manual allocation).
  • Chemical preparation: Clarify if dilution of NaOCl was verified by concentration measurement.
  • Temperature recording: Provide exact location and method of temperature measurement.
  1. Figures and Tables

Issues:
Figures 2–3 lack detailed legends and axis labels. Table 2 lacks indication of statistical grouping method.

  1. Discussion

Issue:
Repetition of results and excessive focus on literature without integrating study limitations.
Suggestion:
Condense repetitive parts, add a paragraph summarizing key limitations (small sample size, absence of material testing, domestic microwave variability), and propose next research steps.

Author Response

We would like to express our sincere gratitude for valuable comments and constructive suggestions. All remarks have been carefully considered, and the manuscript has been revised accordingly.

Comment 1: The title is long and repetitive. I Suggest simplifying it.
Response 1: The title has been simplified and redundant phrases removed, as suggested.

Comment 2: The abstract is verbose and lacks numerical outcomes.
Response 2: The abstract has been condensed, and main quantitative results were added. Redundant phrases have been removed.

Comment 3: Introduction. Well-referenced but overly descriptive; lacks a clear articulation of the research gap.
Response 3: The introduction has been shortened and a final paragraph added that clearly defines the knowledge gap addressed by this study (page 3, lines 104-108)

Comment 4:

  1. Randomization: Specify how samples were randomly assigned to groups (e.g., computer-generated or manual allocation).
  2. Chemical preparation: Clarify if dilution of NaOCl was verified by concentration measurement.
  3. Temperature recording: Provide exact location and method of temperature measurement.

Response 4:

  1. Randomization: Samples were randomly assigned manually. Specimens were not numbered but mixed and randomly packed into sets of five, and for each experimental trial (page 3, lines 129-130)
  2. Chemical preparation: Details on NaOCl dilution verification have been added (page 4, lines 148-149)
  3. Temperature measurement: The method and location of temperature measurement are now clearly specified (page 5, lines169-170)

Comment 5: Figures and Tables. Figures 2–3 lack detailed legends and axis labels. Table 2 lacks indication of statistical grouping method.

Response 5: Figure 2 was improved with axis labels, and more extensive legend. Figure 3 now includes group labels directly on the image, and more extensive legend. Table 2 now specifies the statistical grouping method used.

Comment 6: Discussion- Repetition of results and excessive focus on literature without integrating study limitations.
Response 6: The Discussion has been condensed to reduce repetition. A paragraph summarizing the main limitations and outlining directions for future research has been added (page 9, lines 323-330).